# Symmetry Perception and Psychedelic Experience

**Alexis D. J. Makin** [1,*], **Marco Roccato** [2], **Elena Karakashevska** [1] , **John Tyson-Carr** [1] and **Marco Bertamini** [2]

[1] Department of Psychological Sciences, University of Liverpool, Liverpool L69 7ZA, UK; e.karakashevska@liverpool.ac.uk (E.K.); hljtyson@liverpool.ac.uk (J.T.-C.)

[2] Department of General Psychology, University of Padova, 35131 Padova, Italy; marco.roccato@phd.unipd.it (M.R.); marco.bertamini@unipd.it (M.B.)

[*] Correspondence: alexis.makin@liverpool.ac.uk

**Abstract:** This review of symmetry perception has six parts. Psychophysical studies have investigated symmetry perception for over 100 years (part 1). Neuroscientific studies on symmetry perception have accumulated in the last 20 years. Functional MRI and EEG experiments have conclusively shown that regular visual arrangements, such as reflectional symmetry, Glass patterns, and the 17 wallpaper groups all activate the extrastriate visual cortex. This activation generates an event-related potential (ERP) called sustained posterior negativity (SPN). SPN amplitude scales with the degree of regularity in the display, and the SPN is generated whether participants attend to symmetry or not (part 2). It is likely that some forms of symmetry are detected automatically, unconsciously, and pre-attentively (part 3). It might be that the brain is hardwired to detect reflectional symmetry (part 4), and this could contribute to its aesthetic appeal (part 5). Visual symmetry and fractal geometry are prominent in hallucinations induced by the psychedelic drug *N*,*N*-dimethyltryptamine (DMT), and visual flicker (part 6). Integrating what we know about symmetry processing with features of induced hallucinations is a new frontier in neuroscience. We propose that the extrastriate cortex can generate aesthetically fascinating symmetrical representations spontaneously, in the absence of external symmetrical stimuli.

**Keywords:** Psychophysics; Neuroscience; Aesthetics; Nature-Nurture; DMT; Flicker-induced hallucinations; Comparative Neuroscience





## 1. Introduction

Perception of visual symmetry has been studied with psychophysical methods for over 100 years [1–4]. The neural response to symmetry has been studied in the last 20 years [5–8]. Symmetry may be important in sexual attraction [9–11], and in art and aesthetics [12]. Visual symmetry is a cue for image segmentation [13]. Symmetry is also prominent in visual hallucinations induced by the powerful psychedelic drug N,N-dimethyltryptamine [14], and visual flicker [15].

Our review integrates these topics. First, we consider psychophysical work on symmetry perception (Part 1). Then we consider how the brain responds to symmetry (Part 2). We argue that symmetry is processed automatically, pre-attentively, and unconsciously (Part 3) and that sensitivity to reflectional symmetry is innate (Part 4). Next, we review the aesthetic significance of symmetry (Part 5). Finally, we review the field of psychedelic phenomenology and Flicker-induced hallucinations (Part 6). Although we aim at integration, there is sufficient independence between parts to allow readers to consult each in isolation.

When people use the word 'symmetry' they usually think of reflectional, mirror symmetry. Many papers use the word symmetry as a synonym for reflection because that is the only type investigated. This review covers other types of symmetry and regularity as well.

### 1.1. Part 1: Psychophysical Work on Symmetry Perception

An arrangement is symmetrical if it remains identical after rigid transformation. The 2D planar symmetries are reflection, glide reflection, rotation, and translation (also referred to as repetition). The famous physicist Ernst Mach [16] noticed that reflectional symmetry is more salient than rotation or translation, even when it has the same number of rigid transformations (Figure 1A). Many subsequent psychophysical studies have confirmed this. Reflectional symmetry can be detected efficiently within a single fixation, particularly when the axis is vertical [17–27]. Symmetry perception is noise tolerant, and people can discriminate which of two imperfect symmetries is the more symmetrical [28]. However, non-linearities in the system may reduce sensitivity to slight departures from perfect symmetry [29,30]. Indeed, while discriminating perfect symmetry from asymmetry is fast and parallel, finding small deviations from perfect symmetry may be slow and serial (a possibility discussed, but not always supported [31–34]).

In a typical psychophysical experiment, participants discriminate between symmetrical and asymmetrical displays. On each trial, they press one button if the display is symmetrical and another button if the display is asymmetrical. The dependent variables are often mean response time and accuracy, although accuracy will often reach the ceiling. Response time is faster for salient kinds of symmetry [35].

A different psychophysical procedure is the two-interval forced choice task (2IFC). Each trial presents two patterns in separate intervals. Participants press one button if interval 1 appeared more regular, and another button if interval 2 appeared more regular. The regularity difference between intervals 1 and 2 can be varied experimentally, and thresholds for accurate discrimination can be determined. Lower thresholds indicate higher regularity sensitivity. Thresholds are thus lower for more salient kinds of regularity [36].

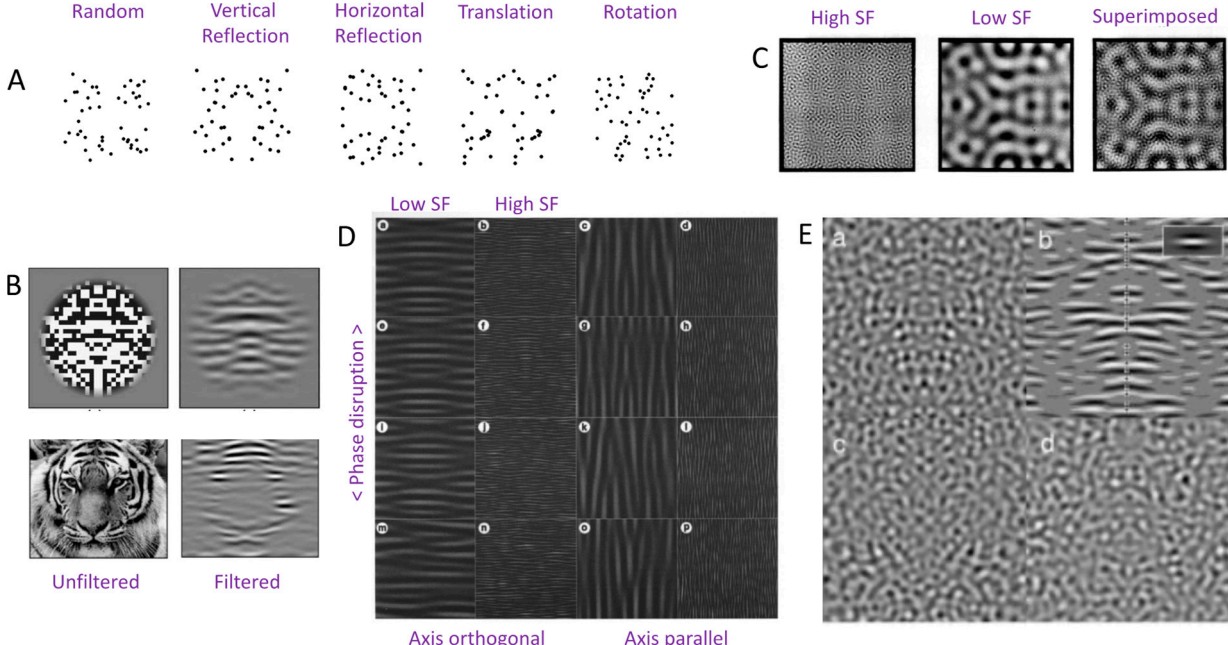

**Figure 1.** (**A**) For most observers, reflectional symmetry is more salient than horizontal reflection, translation, or rotation. (**B**) Filtering images with reflectional symmetry produces mid-point collinear blobs, and the degree of reflectional symmetry is proportional to the degree of blob alignment (examples adapted from Mancini et al. [37] and Rainville and Kingdom, [38]). (**C**) When high and low spatial frequency filtered symmetrical images with orthogonal axes are superimposed, the low-resolution one dominates the percept (from Julesz and Chang, [39]) (**D**) Figure 4 from Dakin and Hess [40]. Phase disruption increases from top to bottom (a–d) 0 deg; (e–h) 60 deg; (i–l) 120 deg; (m–p) 360 deg. Axis orthogonal filtering is shown in the left two columns and axis parallel filtering is shown in the right two columns. Low-frequency filters are shown in columns 1 and 3. High-frequency filters

are shown in columns 2 and 4. Axis orthogonal filtering is more robust to phase disruption (compare i and k for the clearest example). These experiments suggest orthogonal blobs that straddle the axis itself are fundamental for detecting reflectional symmetry. (**E**) Figure 1 from Dakin and Herbert [40]. The original Figure legend reads: *"(a) Spatially band—limited symmetrical texture. (b) Same texture convolved with a low-frequency, horizontally orientated filter (inset), and 'thresholded' to remove values close to the mean grey level. Note the clustering of resulting blobs around the axis and the co-alignment of their centroids (crosses). (c) Symmetrical pattern where the contrast polarity of symmetrical features has been reversed across the axis. (d) Noise pattern containing a strip of symmetry around the axis; at brief presentation times, the entire pattern appears symmetrical."* All examples from published work were reproduced with permission.

### 1.1.1. Filter Models of Symmetry Perception

Rainville and Kingdom [38] urged symmetry researchers to consider the known spatial filtering operations implemented by low-level vision. Low-level vision can be modeled as a retinotopic array of orientation and spatial frequency-tuned filters [41]. Any image can be filtered to yield a frequency-band-restricted version. Filtering can be carried out artificially with image processing software, but something similar is achieved by neural filters in the visual cortex. Image filtering thus isolates stimulus dimensions that are relevant to the visual system.

This approach was taken by the influential blob alignment model of Dakin and Watt [42]. Band pass filtering of an image with reflectional symmetry yields midpoint colinear blobs, orthogonal to the axis. Symmetry detection then becomes a matter of estimating blob alignment in filtered images. Crucially, this mechanism can be just as effective at detecting symmetry in abstract patterns and naturalistic photographs, Figure 1B [37,38].

Images can be digitally manipulated to isolate perceptually relevant spatial frequency components. One early study using this approach found that when horizontal and vertical reflections with similar frequencies are superimposed, the result appears random. However, when their spatial frequencies are more than two octaves apart, they become perceptually separated, although the low band array tends to dominate, Figure 1C [39]. After filtering, stimuli can then be subjected to phase disruption, where the phase of the components is jittered. Beyond a certain point, phase disruption makes symmetry invisible. Perceptually stronger symmetries are more resistant to phase disruption [43]. These experimental approaches support several robust conclusions. The effect of phase disruption is similar for low and high-frequency displays. However, axis-parallel filtered patterns are more fragile and sensitive to phase disruption than axis-orthogonal filtered patterns, Figure 1D [43]. There is an Integration Region (IR), elongated along the axis, where phase disruption disproportionately elevates thresholds. The size and shape of IR are possibly determined by the dominant spatial frequencies in the stimulus. In general, the IR captures about seven cycles of filter output in the y-direction (along the axis) and about 3.5 cycles in the x-direction (straddling the axis). For a $10 \times 10$-degree symmetrical pattern with a spatial frequency of 2.26 cycles per degree, over 95% of the pattern is redundant, Figure 1E [40].

There is some indirect evidence that the avian visual system also has an IR. Birds use symmetry to break moth camouflage, and moths might avoid predation by evolving high-contrast symmetrical patches some distance away from their body midline [44].

The flexible IR mechanism facilitates symmetry discrimination across a range of stimulus sizes, densities, and spatial frequencies. However, there are limits to this flexibility. Increasing the gap between reflected sides can reduce symmetry sensitivity, although this is reversed in patients with macular degeneration [45]. Properties of the IR may also change with eccentricity, even if scaling factors are used to offset declining acuity [46–48].

Several fundamentally distinct filter models of symmetry perception have been proposed [42,49–52], along with related accounts of Glass pattern perception [53]. Despite the differences, all involve taking an image as input, applying a spatial frequency filter, pooling filter responses, and finally, estimating symmetry energy or axis location. Filter-models are

often precisely specified so that researchers can run computer simulations and compare them against human performance.

1.1.2. Findings That Cannot Be Explained by Filter Models

There are several facts about symmetry perception that cannot be directly explained by filter models. Some of these can be accommodated by adding front-end adjustments. One example is the case of *anti-symmetry* (Figure 2A). In a typical experimental stimulus display, reflected elements are of the same luminance: A black (or white) element to the left of the axis is paired with a black (or white) element to the right. Conversely, in one form of anti-symmetry, luminance is mismatched on either side of the axis: A black (white) element on the left is paired with a white (black) element on the right [54–58]. The most basic filter models suggest anti-symmetry should be invisible. However, if filtering is based on rectified luminance information or second-order contrast information, blob alignment still works for anti-symmetry.

Symmetry and anti-symmetry detection is almost equivalent when element density is low [54]. However, anti-symmetry detection selectively declines when density is high, when stimuli are presented in the visual periphery, or when multiple greyscales are involved [37]. Based on these results, Mancini et al. [37] argued that anti-symmetry detection requires a serial visual search for matching positional tokens, while symmetry detection happens automatically and in parallel. However, more recent work suggests the brain responses to symmetry and anti-symmetry can be equivalent in terms of automaticity [59]. Such discrepancies can be explained by stimulus differences. It is likely that different kinds of anti-symmetry are detected in different ways. High-density anti-symmetry may be detected by serial visual search, as proposed by Mancini et al. [37], while low-density anti-symmetry may be detected using second-order contrast information, as proposed by Tyler and Hardage [57] and again more recently by Bellagarda et al. [60].

While filter models can be front-end upgraded to cope with anti-symmetry, there are other fundamental facts about symmetry perception that are not explained by filter models. Firstly, symmetry perception interacts with other gestalt grouping principles. For instance, reflection is more salient when it is a property of a single object, while translation is more salient when it is a property of a gap between objects [35,61–64]. This well-replicated interaction between regularity type and objecthood is not explained by filter models.

Gestalt psychologists observed that regions with reflectional symmetry are often perceived as a figure that has a distinct shape; asymmetrical regions are often perceived as ground that remains shapeless [13,65,66]. Filter models only explain how symmetrical regions are discovered in 2D images, not how they bias border ownership and support figure-ground segmentation. Depth segmentation also interacts with symmetry perception. Reflectional symmetry is harder to detect if the left and right sides are presented on different depth planes so that one side appears closer and the other further away [67]. This finding is intuitive, but again not explained by filter models. Perhaps filtering operations only happen within disparity-defined depth planes?

The next complication is that symmetry perception works with substructures and parts that are themselves smaller gestalts. Locher and Wagemans [68] note that local substructures on each side of the axis often serve as 'input primitives' in the construction of the global percept. Convexities and concavities may be input primitives when detecting symmetrical polygons [69,70]. Filter models do not give special status to substructures, convexities, or concavities.

Perhaps most importantly, filter models only directly explain the detection of 2D symmetry in the frontoparallel plane. This is a substantial failure of ecological validity because symmetrical 3D objects rarely project symmetrical 2D images under naturalistic viewing conditions (Figure 2C). However, the brain can overcome perspective distortion and detect symmetry in objects quite efficiently [71–75]. There is usually a performance cost when detecting reflectional symmetry from non-orthogonal perspective viewpoints, but this cost is not large enough to imply symmetry detection has simply failed [76]. Indeed,

slanted symmetry detection can match frontoparallel symmetry detection when binocular disparity information aids perception of surface slant [77]. Filter models do not address such perceptual interdependencies, where the outcome of one perceptual operation (e.g., view invariance) determines the outcome of another (e.g., symmetry perception).

While view-invariant representation aids symmetry detection, an axis of symmetry can aid view-invariant representation [78–82]. There are various ways this bidirectional relationship could manifest. For example, the visual system may assume objects are symmetrical, particularly if they are familiar, and back-calculate view-angle from retinal distortion. The resulting view angle information could then support inferences about the shape of *other* objects in the same scene. Likewise, objects with familiar sizes (such as playing cards) allow us to back-calculate viewing distance from retinal size. These objects are then reference points that facilitate inferences about the size of all other objects in the scene.

Symmetry may also aid view invariance in a second, and very different, sense. Children often mistake lateral mirror image letters (b and d) but not vertical mirror image letters (b and p). There are at least three many empirical demonstrations of this subjective left-right interchangeability. First, in visual search tasks [83], mirror-image distractor lines (e.g., / and \) appear similar, and this causes targets with a third orientation to pop out [84]. Second, left-right reversed Japanese films can be enjoyed without English readers noticing a change, while turning the screen upside-down would be absurd and immediately obvious [85]. Third, neural responses to mirror image stimuli are very similar in the monkey inferotemporal cortex [86]. This ubiquitous perceptual indifference to mirror reversal may be an intermediate stage in achieving complete view invariance [87]. These papers claim that symmetry supports view invariance, but this is one member of a family of loosely related phenomena.

Not all studies on perspective symmetry have used optically realistic stimuli [88]. Ideally, the location of the virtual camera and the participant's eye should match, giving a superior *perspective transformation*. Without this eye-camera match, we are left with an inferior *projective transformation*. With the inferior projective transformation, observers must make two corrections, first adopting the standpoint of the virtual camera, and then correcting perspective distortions introduced by non-orthogonal camera angles. Perspective transformation is easily achieved if participants view stimuli printed on a physical board, as in [71]. However, perspective transformation is rarely simulated on a 2D computer screen. The use of projective transformations [72,89] may lead researchers to *overestimate* the costs of perspective distortion.

Perspective transformations are preferable, but they introduce a second complication: Symmetry distortion around the axis may be minimal, and therefore retinal symmetry in the all-important IR can be almost intact despite large changes in viewpoint, as in [71]. This can lead researchers to underestimate the costs of perspective distortion. This second problem can be overcome by introducing both slant and tilt, which substantially distorts image symmetry in the IR.

Filter models only work with perspective symmetry if low-level vision can be loaded with a mentally rotated version of the symmetrical stimulus in a top-down fashion. Much recent work has shown that the top-loading of V1 is ubiquitous, so this account is quite plausible [90]. Additional features, such as conspicuous square frames or perspective lines, might facilitate mental rotation and top loading of V1.

Even when optically realistic stimuli are used, there is some uncertainty about how 3D symmetry is processed. The brain may construct view-invariant representations of symmetry under some conditions (these can also be referred to as object-level, allocentric, or post-constancy representations). However, the visual system might be able to avoid this computationally costly operation by using optic invariants instead [91]. For instance, if symmetry lines converge on a vanishing point, then planar symmetry is present in the distal object. The brain may simply detect convergence on a common vanishing point, and

then behave as if symmetry is present in the world, without constructing an object-level representation of planar symmetry [88,92].

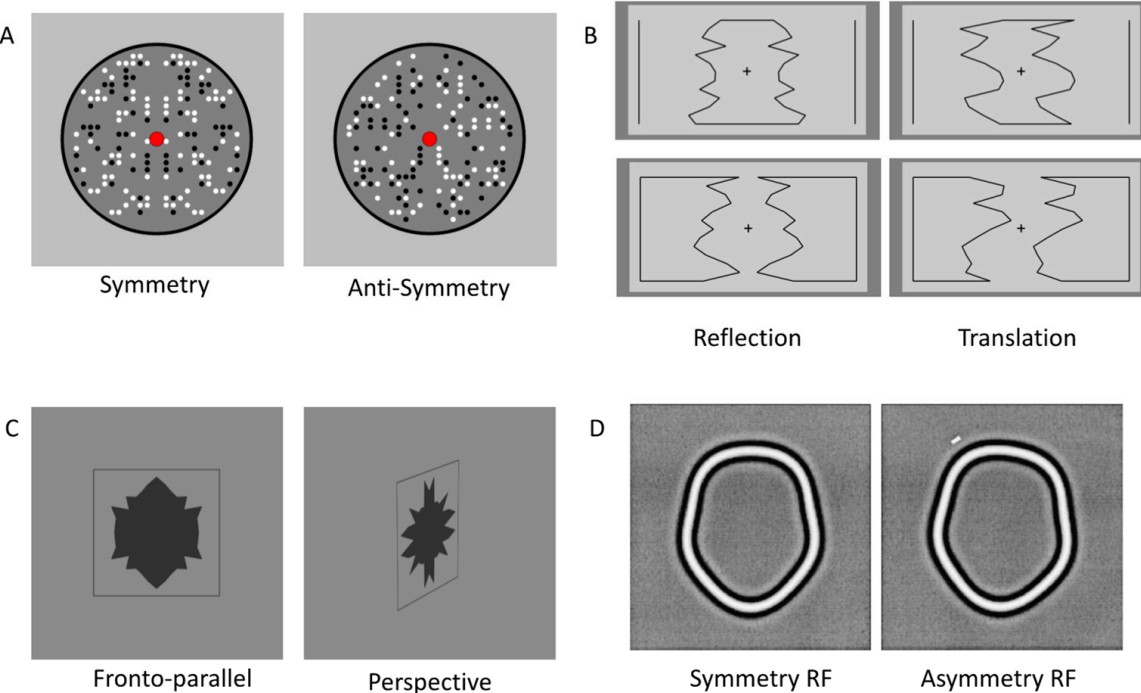

**Figure 2.** (**A**) Anti-symmetry perception requires modification of filter models with front-end rectification of or second-order signals. (**B**) Reflection is more salient when it is the property of a single object. Translation is more salient when it is the property of a gap between two objects. This interaction between regularity type and objecthood is not explained by filter models. (**C**) Most experiments use symmetry in the frontoparallel plane. In the real world, symmetrical objects are often seen in perspective and do not project a symmetrical image onto the retina. This is a complication for filter models. (**D**) Symmetry in radial frequency patterns might be a special category (Adapted with permission from Wilson and Wilkinson, [93]).

### 1.1.3. The Bootstrapping Model

An alternative to filter models of reflectional symmetry perception is the *bootstrapping model* [34,73,94] which builds on earlier ideas from Jenkins [95]. The bootstrapping model states that reflectional symmetry detection is facilitated by the presence of second-order structures, such as symmetrical quadrangles (e.g., rectangles and trapezoids) made of virtual lines between four elements. It is advantageous when the midpoint of a symmetrical quadrangle suggests a direction of search for the next symmetrical quadrangle. Quadrangle discovery can then spread rapidly along the global axis of reflection (bootstrapping). Skewing destroys quadrangle alignment in single reflections and impairs symmetry detection. Bellagarda et al. [96] provided further support for the bootstrapping model using Gabor-corners as elements rather than dots. When Gabor-corners lay on top of the corner of virtual quadrangles, symmetry detection was enhanced. Bellagarda et al. [96] noted that blob-alignment type filter models do not account for the facilitatory role of second-order structure in symmetry detection. Dry [97] provided a mathematically sophisticated extension of the bootstrapping model, called the Voronoi tessellation model. He found that it provided a better fit to existing psychophysical data than the four filter models in Dakin and Watt [42]. Although the bootstrapping model does not explicitly discuss the role of visual attention or eye movements, studies suggest that people spontaneously move their eyes up and down the axis *as if* they are searching for midpoint colinear quadrangles [98–100], although see [101,102]. However, axis scanning is unlikely to be a behavioral manifestation of the bootstrapping operation because it happens over a much longer interval.

### 1.1.4. Symmetry in Radial Frequency Contours

Closed radial frequency contours can be drawn by adjusting the radius as we move round the perimeter of a circle. Imagine that a motor slowly turns a compass to draw a circle, while a second motor moves the arms in and out with a sinusoidal oscillation. If motor two runs through five cycles for every turn of motor one, the machine draws a smoothly rounded five-pointed star. The phase of oscillation determines the orientation of the star. Tiny radial frequency modulations can be discriminated from circles in the hyperacuity range, suggesting special mechanisms are involved [103]. Visual adaptation studies show that radial frequency code is contrast and size invariant [104]. Radial frequency contours might help with discriminating head shape or provide mid-level shape primitives to face and object-sensitive regions further up the ventral stream. Wilson and Wilkinson [93] found that symmetry discrimination of radial frequency contours was extraordinarily good. They concluded that:

> " . . . *results all point to the presence of two different symmetry detection mechanisms in human vision. The first is involved in processing the symmetry of textured surfaces exemplified by random dot patterns, and plausible models of the underlying neural mechanisms have been suggested (Dakin and Hess, 1997; Dakin and Watt, 1994). Based upon our data, a second bilateral symmetry mechanism is optimized for processing the symmetric contours defining many biological shapes".*

### 1.1.5. Symmetry in Optic Flow Fields

As we move around, the whole retinal image shifts in the opposite direction, creating optic flow fields. Motion patterns in optic flow fields are a crucial source of information for rapid dorsal-stream-mediated action control. The radial symmetry of velocity signals in an expanding optic flow field tells us we are heading straight toward the static point in the center (pilots may use this information when aiming for the runway). Different optic flow fields are coded in motion-sensitive region MST [105].

### 1.1.6. Summary of Part 1

In summary, filter models explain how we detect symmetry in the same way across a variety of images. However, they do not directly explain interactions between symmetry and other forms of perceptual organization and do not generalize well to 3D objects. Filter, bootstrapping, and optic invariant models should be understood as applications of different theoretical traditions to the problem of symmetry perception. Symmetry in closed radial frequency contours and optic flow fields may be computed in different ways. In part 2, we consider the brain response to visual symmetry. Most things that enhance symmetry detection in psychophysical tasks also enhance the neural response.

### 1.2. Part 2: How the Brain Processes Symmetry

### 1.2.1. fMRI

The neuroscience of symmetry perception was reviewed by Bertamini et al. [6]. Here, we update this review with developments from the last five years. Functional MRI studies usually present participants with symmetrical or asymmetrical dot patterns. Symmetry activates a network of brain regions in the extrastriate cortex [106,107]. The strongest symmetry responses are in early ventral stream regions, particularly V4, and in the shape-sensitive lateral occipital complex (LOC). This role of the LOC in symmetry perception is unsurprising. In many fMRI studies, the LOC is functionally localized by comparing objects to scrambled objects. The LOC codes visually meaningful shapes and gestalt [108,109] and symmetry is a visually meaningful gestalt.

This extrastriate symmetry response has now been replicated across a range of stimuli and tasks [110–112]. More recent fMRI work recognizes more extrastriate visual regions, but symmetry activates all of them in approximately the same way [111,112]. The extrastriate symmetry response has been replicated in monkeys viewing regular wallpaper patterns [113] and in humans using functional Near Infrared Spectroscopy fNIRS, [114].

As well as replicating the univariate extrastriate symmetry response, van Meel et al. [115] used multivoxel pattern analysis (MVPA) and found that symmetry information could be decoded from voxels of the extrastriate cortex. Decoding the difference between symmetry and asymmetry was superior to decoding the difference between (a) two different symmetries or (b) two different asymmetries. Furthermore, having one half in common had no cost for symmetry vs. asymmetry decoding in LOC: This suggests the LOC codes the global configuration and is indifferent to local features. In contrast, having one half in commonly reduced symmetry vs. asymmetry decoding performance in V1: This suggests V1 is sensitive to local features and is indifferent to the global configuration. Although V1 is not apparently activated by symmetry, global axis orientation information might be coded in V1 based on top-down signals [116].

### 1.2.2. TMS

Experiments with Transcranial Magnetic Stimulation (TMS) have shown that LOC disruption selectively impairs symmetry discrimination [5,117–119]. This effect is maximal between 130 and 250 ms post-stimulus onset [120]. The right hemisphere specialization suggested by some TMS studies is consistent with psychophysical work, which also finds that the right hemisphere is more symmetry sensitive than the left [121,122].

### 1.2.3. Invasive Recordings

When reviewing the fMRI results of Sasaki et al. [106], Beck et al. [123] asked whether symmetry was coded by single neurons within the extrastriate cortex, or whether the symmetry response is best captured by a sophisticated population code that can be revealed by lower resolution methods such as fMRI. There is some evidence for the single neuron account. First, we note that reflectional symmetry can be detected on the visual forward sweep [124]. This implies that symmetry-grandmother cells are triggered before recurrent processing mediates perceptual grouping. Second, two influential papers [125,126] reported that V4 cells have preferences for either radial gratings (with rotational symmetry) or hyperbolic gratings (with multiple axes of reflectional symmetry). This suggests single cells in V4 fire in response to specific types of symmetry. More recently, McMahon and Olson [127] found that inferior temporal cortex (IT) neurons responded more strongly to symmetrical than asymmetrical shapes (although this was not the main purpose of their study). There is a risk of false dichotomy when reading Beck's distinction. Even if there are symmetry-grandmother cells in the extrastriate cortex, it is likely that sophisticated population codes are also involved.

Pramod and Arun [128] presented monkeys with asymmetrical shapes (with two different halves) or symmetrical shapes (with the same half mirrored) while recording from IT neurons. They found that the response to symmetry was the sum of the responses to the halves presented in isolation. This meant two symmetrical objects were more distinct from each other than two asymmetrical objects because each part is given twice in a symmetrical object. Pramod and Arun [128] provide an analogy:

> *"Just as mixing diverse paints produces a homogeneous overall colour, adding heterogeneous parts within an asymmetric object renders it indistinct. In contrast, adding identical parts within a symmetric object renders it distinct."*

Pramod and Arun [128] suggest that the standard account of extrastriate symmetry sensitivity is wrong. Instead, they propose that symmetry may be coded as neural distinctiveness. Some results may be reinterpreted as the distinct symmetrical stimuli breaking ubiquitous fMRI adaptation.

While Pramod and Arun's work is an excellent contribution, we would argue that their distinctiveness account is less plausible than the standard account. Reflectional symmetry produces a stronger extrastriate response than translation, even though reflection and translation both give two doses of the half. The spatial relationship between the sides really matters in symmetry perception [16,106,129]. Furthermore, van Meel et al. [115] found that configural properties determine LOC response in humans.

### 1.2.4. Neuropsychological Studies

Ventral stream lesions produce object agnosia, sometimes including symmetry blindness [130]. Meanwhile, dorsal stream lesions produce hemispatial neglect [131]. Most commonly, right parietal lesions cause patients spontaneously to ignore the left side of objects (and not simply the left visual hemifield). Driver et al. [132] found that left hemispatial neglect patient CC could not consciously judge whether vertical symmetry was present or not because one side of the stimulus was subjectively absent. However, CC still processed symmetrical regions unconsciously and interpreted them as figures. Another neuropsychological study on patient JW provides a double dissociation. JW had impaired perceptual grouping (ventral) but normal visuospatial attention (dorsal). Unlike CC, symmetry did not help JW with figure-ground segmentation [133].

### 1.2.5. Sensitivity to Haptic Symmetry

The word 'haptics' refers to our faculty for identifying objects in the absence of vision by active, exploratory touch. People have a remarkable ability to haptically identify objects with their hands in the dark. They can also haptically determine whether an object is symmetrical or not [134,135]. Haptic symmetry activates the LOC in early blind patients [135]. Studies on early and late blind patients suggest visual experience is required for the development of some aspects of haptic symmetry detection [136].

### 1.2.6. EEG

Following early work by Beh and Latimer [137], the extrastriate symmetry response has been extensively investigated with EEG. Symmetrical and asymmetrical stimuli both produce Event-Related Potentials (ERP) at posterior electrodes. After the P1 and N1 components of the visual evoked potential, the amplitude is lower for symmetrical stimuli [138–140]. This difference wave is often called the *Sustained Posterior Negativity* (SPN). SPN amplitude scales with the proportion of symmetry in symmetry + noise displays (a variable called Psymm, Figure 3, [141,142]). Source localization suggests the SPN is generated by two dipoles in the bilateral extrastriate symmetry network [143,144]. However, presenting perfect symmetry to just one hemifield generates a contralateral SPN over the opposite hemisphere [145], although low Psymm may only be detected at fixation [46].

### 1.2.7. Bottom-Up Effects on SPN Amplitude

SPN amplitude is largely determined by stimulus features. For instance, 2-fold symmetry generates a larger (i.e., more negative) SPN than 1-fold symmetry [146], and the SPN is larger for reflection than rotation or translation [129]. In contrast, SPN amplitude is not substantially influenced by the nature of the elements that carry these symmetrical relationships. A similar SPN is generated by isochromatic (luminance-defined) and isoluminant (color-defined) symmetry [147]. Likewise, a similar SPN is generated by contrast- and disparity-defined symmetrical polygons [148]. We even obtain a similar SPN for symmetrical abstract patterns, flowers, and landscapes [149]. This is consistent with behavioral data: any elements can support symmetry perception if they are matched at symmetrical locations [19]: Symmetry can be detected in color- and luminance-defined elements [36] orientated lines [68] or oriented Gabors [150].

SPN amplitude scales with the perceptual goodness of different regularities. Perceptual goodness is a term from Gestalt psychology. It refers to the perceptual strength or obviousness of a configuration. Perceptual goodness can be coded by the W-load metric from the *holographic weight of evidence model* [151]. The holographic model states that $W = E/N$, where W is perceptual goodness, E is the number of "holographic identities", and N is the number of elements. A holographic identity is a local substructure with the same regularity as the global whole. For instance, one pair of reflected dots is a holographic identity in a dot pattern with global reflectional symmetry. A holographic regularity has a whole number of holographic identities, with no remainder. For dot patterns with 1-fold reflectional symmetry, $W = 1/2$, because every pair (E) has two dots (N). The holographic

model predicts both subjective differences in perceptual goodness and psychophysical results [152,153]. Makin et al. [146] found that W explained over 80% of the variance in grand average SPN amplitude. Most remarkably, Experiment 3 found that the subtle non-monotonic relationship between W and the number of folds [154] was captured by SPN amplitude in an early window (although this finding requires replication with greater variation of orientations). Although the W-load metric is well supported by SPN research, it may be that alternative models [155,156] can predict more variance in SPN amplitude than W. We see this as a challenge for future research.

Given the number of studies now completed, we can confidently predict that any stimulus manipulation that substantially alters symmetry salience will alter SPN amplitude. The psychophysical literature is thus a rich source of testable predictions. For instance, axis parallel filtering is likely to result in a weaker SPN than axis orthogonal filtering [43]. Phase disruption inside the IR is likely to reduce SPN amplitude, while phase disruption outside the IR will have no effect [38]. Presenting left and right sides on different depth planes is likely to reduce SPN amplitude, even if the enantiomorphs are the same retinal size [67].

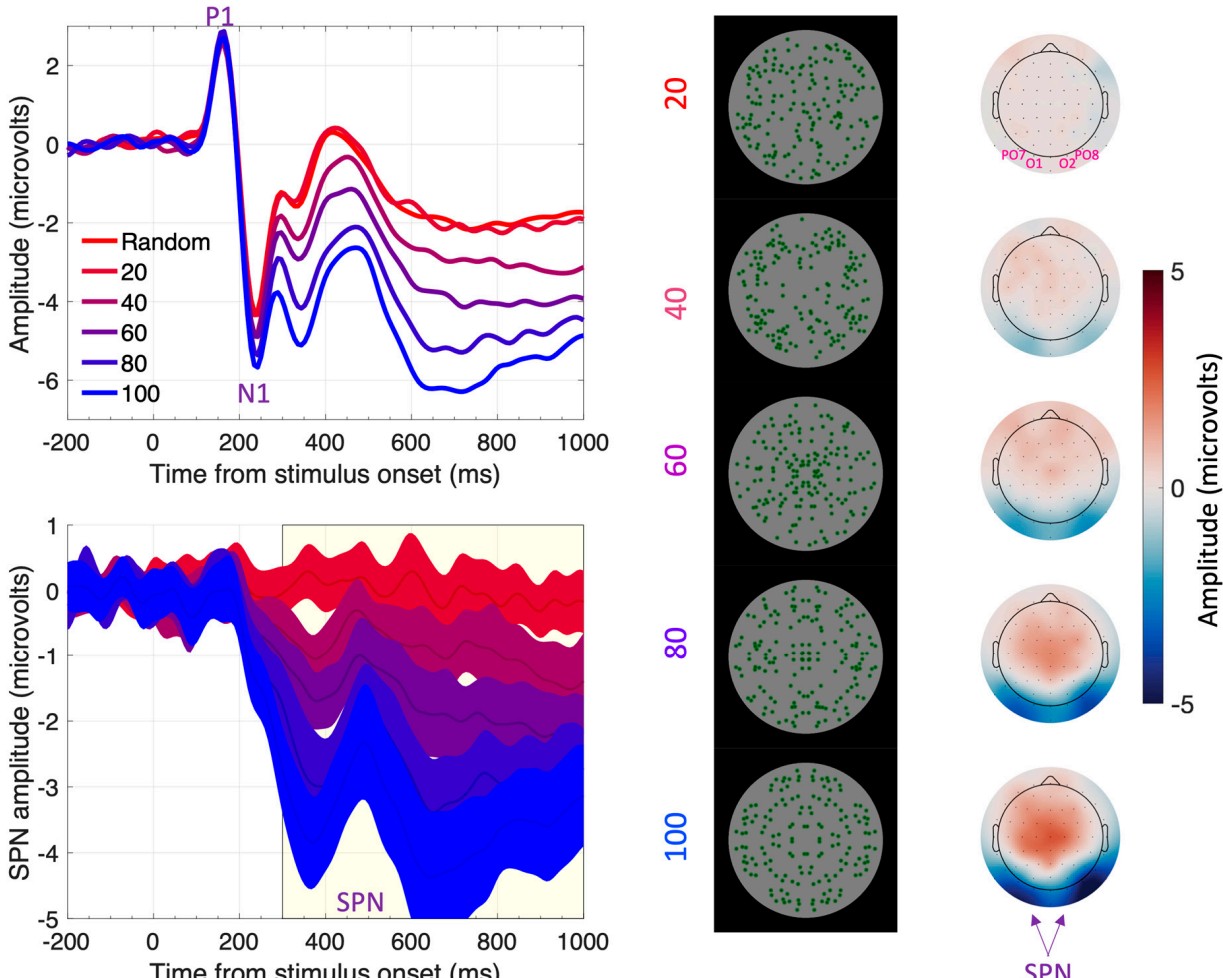

**Figure 3.** (from Makin et al. [7]) The grand-average ERPs are shown in the upper left panel and difference waves (reflection-random) are shown in the lower left panel. A large SPN is a difference wave that falls a long way below zero. Topographic difference maps are shown on the right, aligned with the representative stimuli (black background). The difference maps depict a head from above, and the SPN appears blue at the back. Purple labels indicate electrodes used for ERP waves [PO7, O1, O2, and PO8]. Note that SPN amplitude increases (that is, becomes more negative) with the proportion of symmetry in the image. In this experiment, the SPN increased from 0 to ~−3.5 microvolts as symmetry increased from 20% to 100%. Adapted from Figures 1, 3 and 4 in Makin et al. [141].

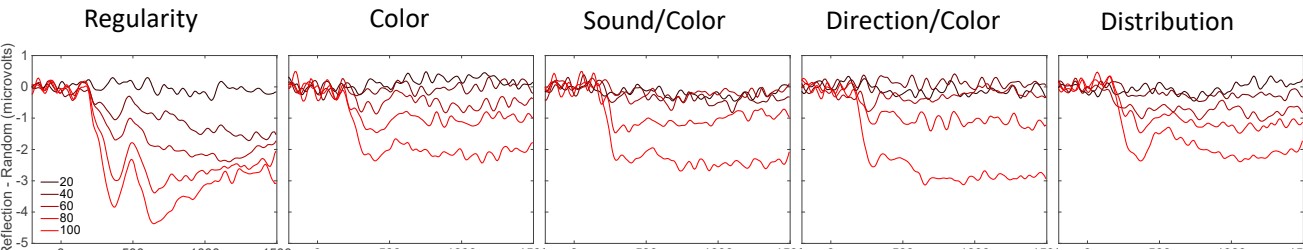

**Figure 4.** Parametric SPN responses in five different tasks. The default parametric response in the four non-regularity tasks (Color, Sound/color, Direction/color, and Distribution) was enhanced in the Regularity task (left panel). N = 26 in each task. Results from Makin et al. [141].

### 1.2.8. Top-Down Effects on SPN Amplitude

The SPN is very robust to experimental manipulations of task, but not completely indifferent to them [149,157–159]. Makin et al. [141] found that the parametric SPN response was comparable across five tasks but selectively enhanced when participants attended to regularity (Figure 4). This conclusion is consistent with the finding that attention boosts all forms of perceptual organization [160,161].

Tyson-Carr et al. [144] reanalyzed the data set of Makin et al. (2020) and found that the left and right hemisphere extrastriate dipoles were joined by a third dipole in the posterior cingulate during the regularity task, but only when the degree of symmetry was 80 and 100%. This previously unknown third symmetry response may reflect binary decision-making rather than perceptual representation (see also [162] for distinctions between perceptual and decision-related activity).

### 1.2.9. The Complete Liverpool SPN Catalogue

A substantial step forward since 2018 has been the development of the complete Liverpool SPN catalog. This is a complete public database of all SPN recordings from the University of Liverpool (https://osf.io/2sncj/, accessed on 28 June 2023). Users of the SPN catalogue on the open science framework may begin with the SPN gallery (https://osf.io/eqhd5/, accessed on 28 June 2023), which currently has one page for each of the 249 grand average SPNs recorded so far.

As explained by Makin et al. [7], the SPN catalog can be used for meta-scientific purposes. There is increasing anxiety about the replicability of published science [163,164]. Makin et al. [7] assessed the extent to which Bishop's "four horsemen of irreproducibility" undermine SPN research. While publication bias, p-hacking, and HARKing were minor problems, low statistical power was identified as a major issue. It would take at least 38 participants to reliably record a 0.5 microvolt SPN or SPN modulation, and our typical sample size is just 24. Low statistical power is a chronic problem in cognitive neuroscience [165], and there is room for improvement in SPN research too.

The complete catalog can also support scientific conclusions that could never be gleaned from a single experiment. Two predictors, W and Task (both coded on a 0–1 scale), explained 33% of the variance in grand average SPN amplitude (SPN (microvolts) = $-1.669$ W $- 0.416$ Task $+ 0.071$, Figure 5). Aggregated analysis also shows that the laws of perceptual organization that determine SPN amplitude are similar in both hemispheres. There is no evidence, for example, that the left hemisphere cares more about Task and the right cares more about W. Crucially, these conclusions are based on analysis of all available SPN data, published and unpublished, and are thus untouched by file drawer problems.

### 1.2.10. Retinal and Extraretinal Symmetry Representations

When considering all SPNs, one sees an important distinction between responses to retinal symmetry and extraretinal symmetry. The brain always responds to retinal symmetry, whatever the participant's task. Under some conditions, the brain can go beyond the image, and recover extra-retinal symmetry in objects despite complications

caused by factors such as perspective distortion and partial occlusion. Brain responses to extraretinal symmetry are more fragile and task-dependent.

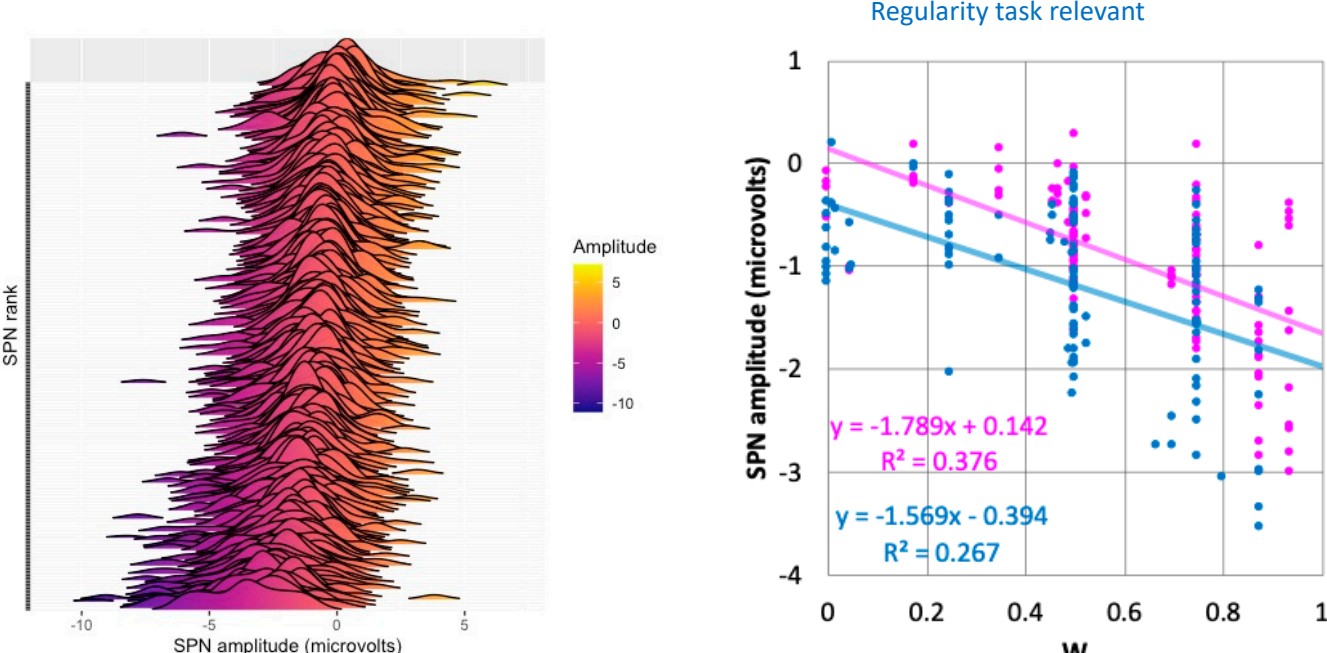

**Figure 5.** The left panel shows a ridge plot of all 249 SPNs currently in the complete Liverpool SPN catalog. Each distribution (ridge) shows the spread of participant-level SPNs around the grand average. Larger (more negative) SPNs are lower on the Y dimension. The right panel shows that variance in grand average SPN amplitude can be partly explained by W-load, and partly by task.

Makin et al. [89] found that SPN was the same for flat and slanted displays when participants were attending to regularity. However, the SPN was selectively reduced for slanted displays when participants were attending to element color. This suggests participants were actively constructing a view-invariant representation during the regularity task. It suggests they were not relying on optic invariants to make their judgment, as proposed by [88]. It is also noteworthy that the SPN onset was not substantially delayed in the regularity task. If mental object rotation preceded symmetry discrimination, it must have been much quicker than the kind of mental object rotation made famous by Shepard and Metzler [166]. This study requires replication with optically realistic stimuli and larger samples. However converging evidence comes from an fMRI study by Keefe et al. [112], and comparable conclusions were reached by Wilson and Farah [167].

Conceptually similar results come from the experiments of Rampone et al. [168]. On each trial, participants viewed a half-occluded polygon for 500 ms. The occluder then shifted to reveal the other half of the polygon for 1000 ms. The whole polygon was either symmetrical or asymmetrical. When the second half of a symmetrical polygon was revealed, an SPN was recorded, despite the fact there was no symmetry present in the retinal image (Figure 6A). This was replicated in two further experiments by Rampone et al. [168]. Rampone et al. [169] also replicated the effect when parts 1 and 2 were not just separated in time, but no longer in the same retinotopic frame (Figure 6B). However, there was no SPN when participants attended to color in Experiment 5 of [168]. These shifting occluder experiments again show extraretinal symmetry representations are only generated when they are task-relevant.

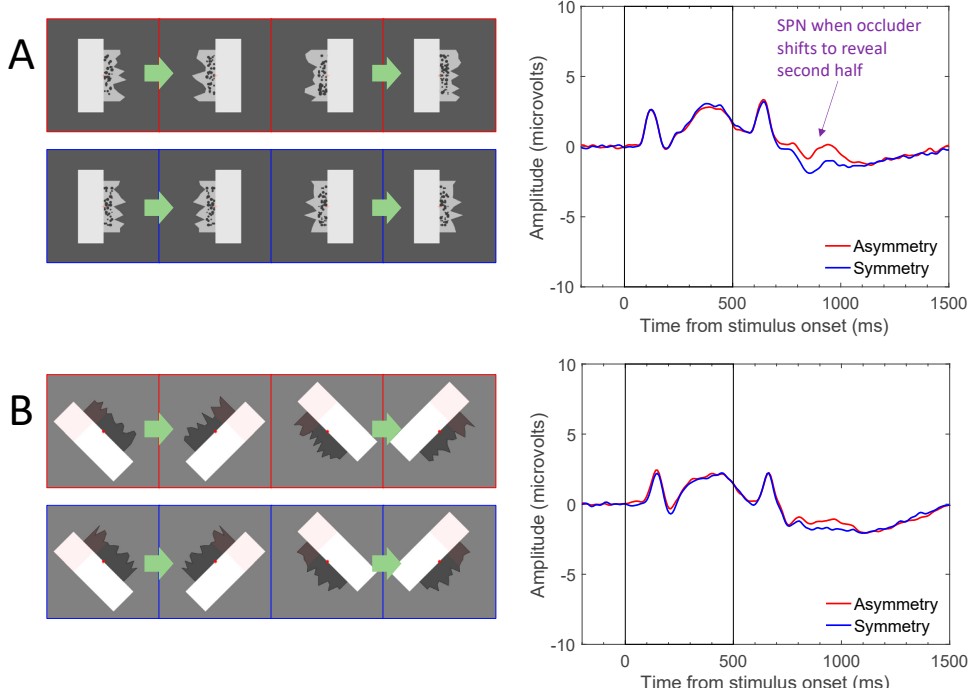

**Figure 6.** Results from shifting occluder experiments of Rampone et al. [168] (**A**) and Rampone et al. [169] (**B**). An SPN is generated once the occluder shifts and reveals the second half of a symmetrical pattern.

Rampone et al. [170] conducted further comparable experiments. Revealing one-half of a previously seen symmetrical or asymmetrical polygon was enough to generate an SPN. This SPN could have been caused by the recall of a polygon stored in visual short-term memory. This memory reactivation SPN was still generated when participants were attending to color in Experiment 3 of Rampone et al. [170]. This is perhaps a rare example of a task-irrelevant extraretinal symmetry-generating SPN. Another example comes from Bertamini et al. [171], who found that the SPN can persist after stimulus offset. This persistent post-stimulus SPN was found even when participants were not attending to regularity. However, this SPN was largely abolished by visual noise masking. It is unclear whether post stimulus persistence without visual masking should be classed as an extraretinal symmetry response or not.

In summary, the evidence suggests that strong retinal symmetry is processed whatever the task, and extraretinal symmetry is more fragile, and perhaps only processed only when it is task-relevant. The next stage for SPN research is to subject these two claims to rigorous testing. First, can we find tasks that make the brain blind to high W retinal symmetry? For instance, this might happen when participants are reading superimposed words of negative valence [172] although preliminary data suggest this is unlikely. Second, can we find conditions where the brain still processes task-irrelevant extraretinal symmetry? This might happen when there are sufficient cues to support 3D interpretation [77].

### 1.2.11. Specificity of the SPN

We have often described the SPN as an index of the extrastriate *symmetry* response. However, we acknowledge that this may be too specific. The SPN is probably generated whenever the extrastriate cortex encodes non-accidental spatial relationships between parts. This may happen with objects compared to scrambled objects [173]. The SPN can also be generated by Glass patterns [174] (Figure 7). Glass patterns are *not* technically a form of symmetry, but they are a form of regularity. Glass pattern detection requires pooling local orientation information to construct a global gestalt [53,175,176]. Like reflectional symmetry, Glass patterns also activate V4 and LOC [177,178]. However, unlike reflection, Glass patterns may not reduce subjective numerosity [179] and Glass patterns have implied

motion that may also activate motion-sensitive V5 [180]. Circular and Radial Glass patterns generate an SPN with an amplitude predicted by the holographic model [146]. However, the SPN response to translational Glass patterns is anomalously low [181].

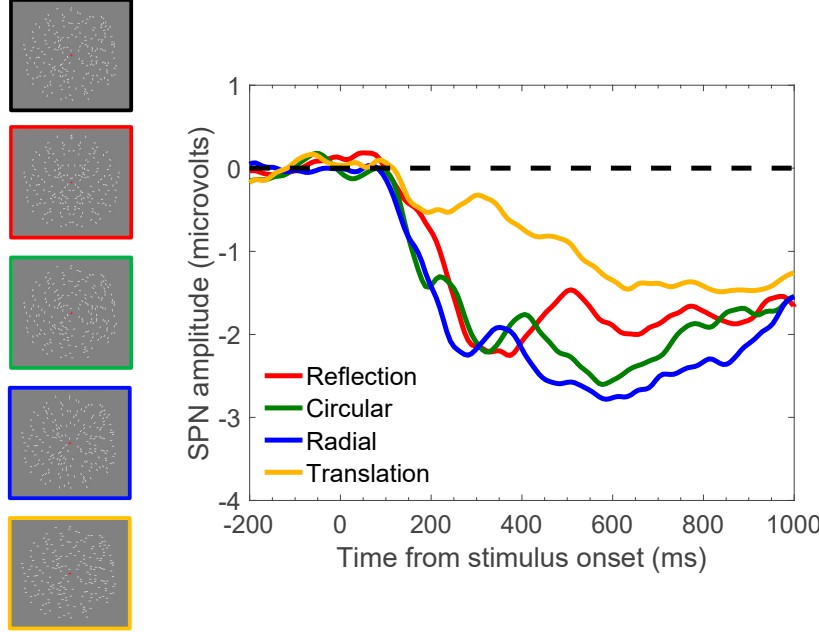

**Figure 7.** The SPN responses to reflectional symmetry and three types of Glass patterns. As with other SPNs, these waves are computed as a difference from a random condition (black square). These results from Rampone and Makin [181] show that reflectional symmetry is not the only regularity that activates the extrastriate cortex and generates an SPN (All waves fall below the dashed black zero line).

Repetition paradigms suggest some functional overlap between different kinds of regularity. Nearly all neural responses change with repeated presentation. If the neural response to stimulus A is altered by the prior presentation of stimulus B, then we can assume they A and B are coded by overlapping neural populations. Recent research shows that the presentation of three different regular patterns in rapid succession increases SPN amplitude. This is known as SPN priming, and it may be related to other known dynamic enhancements [182,183]. SPN priming does not transfer between hemispheres, or between reflections with unpredictably changing orientations. However, it does transfer between horizontal and vertical reflection, and between vertical reflection and 90-degree rotation [159]. Furthermore, SPN priming also transfers from black to white exemplars [184], and between symmetry and Glass patterns [185]. In these papers, we noted that the transfer of SPN priming is more liberal than the filter models allow.

### 1.2.12. Steady-State Visual Evoked Potentials

Norcia et al. [186] measured symmetry responses with the steady-state visual evoked potential (SSVEP) approach [187]. Alternating sequences of reflection and random dot patterns can be presented at 2 Hz, and this repetitive stimulation drives a strong 2 Hz oscillation in the brain. The symmetry response is captured by the odd harmonics of this oscillation. Inverse Fourier transformation of the odd harmonic reveals a wave a lot like the SPN, beginning at about 220 ms and then sustained until after the stimulus switches. Like the SPN, the odd harmonic response scales with the number of axes of reflection and is stronger for symmetry than anti-symmetry [188]. Kohler et al. [111] found that the odd harmonic response codes the order of rotation symmetry in regular patterns known as wallpaper groups (using a subset of groups codenamed P2, P3, P4, and P6). Other work with SSVEP has further characterized the steady-state response to

symmetry [189] Most recently, Kohler and Clarke [190] reported odd harmonic responses to symmetries in all 16 wallpaper groups alternating with the least regular group, P1. This scaled with the subjective salience of the wallpapers measured using a symmetry detection task. Sophisticated source analysis identified an early onset in brain areas V3d (75 ms) and V4 (77 471 ms), which preceded responses in LOC (110 ms) [111].

### 1.2.13. Alpha Desynchronization

Rhythmical oscillations are ubiquitous in the brain, even the in absence of periodic stimulation. They are caused by populations of neurons alternating between periods of high and low excitability. Like ocean waves, brain waves show an inverse relationship between amplitude and frequency: Low-frequency waves have the highest amplitude and vice versa. However, alpha oscillations (approximately 10 Hz) stand out above the amplitude-frequency slope. The occipital alpha rhythm is prominent at posterior electrodes when participants close their eyes, and opening the eyes leads to alpha desynchronization [191]. Alpha desynchronization reflects activation of the visual cortex. The traditional assumption was that alpha reflects cortical idling. It is now widely supposed that alpha mediates executive control, timing, and top-down inhibition as well [192].

All visual onsets produce event related alpha desynchronization (Alpha ERD). Alpha ERD is far less sensitive to symmetry manipulations than the SPN, but some systematic effects can be discerned with large samples. Two early studies suggested alpha ERD was (1) unaffected by stimulus regularity, and (2) selectively enhanced over the right hemisphere when regularity is task-relevant [193,194]. A new time-frequency analysis of the dataset from Makin et al. [141] suggests both these claims are both incorrect. Pre-stimulus alpha is stronger over the right hemisphere by default [195]. After stimulus onset, alpha power reduces and equalizes over both hemispheres. Therefore, the change from the pre-stimulus baseline is stronger over the right hemisphere. Furthermore, like the SPN, alpha ERD increases with the proportion of symmetry in the image, and this happens in all tasks. Interestingly, the relationship between Psymm and alpha power is stronger over the right hemisphere, and this is not an artifact caused by differential baseline alpha power [196].

## 2. Part 2 Summary

Many non-accidental visual configurations activate the shape-sensitive LOC and other parts of the extrastriate cortex. Symmetrical arrangements produce strong responses. The extrastriate symmetry scales with the goodness of symmetry in the image, as shown with fMRI, the SPN, SSVEPs, and alpha ERD. Anything that alters symmetry discrimination performance is likely to alter SPN amplitude. There is strong evidence that retinal symmetry activates the extrastriate cortex automatically. However, this requires careful consideration in part 3.

## 3. Part 3: Symmetry Processing Can Be Automatic, Pre-Attentive, and Unconscious

Psychologists often talk about stimuli as being processed automatically (or not), pre-attentively (or not), and unconsciously (or not). How much visual processing happens automatically? How much visual processing happens in the absence of attention? How much happens unconsciously? These are classic but difficult questions, that are inter-linked [124,197–200]. These are also standard topics in symmetry perception research, but they require careful treatment. Researchers risk using the terms interchangeably while glossing over important distinctions.

One difficulty arises from causal relationships: A stimulus can be first processed automatically, *causing* it to grab visual attention, *causing* to enter conscious awareness. These causal steps are gated by numerous internal and external variables. A second difficulty is objectively determining what happened in the brain on a given trial. A stimulus may be irrelevant to the primary task given to the participant, but they may spontaneously attend to it and categorize it anyway if processing resources are available.

It is particularly difficult to determine whether a stimulus was processed consciously or not. All we have is the participants' verbal report, but an accurate verbal report depends on other cognitive operations, which may themselves fail. It is always possible that participants were briefly conscious of a stimulus, but then rapidly forgot about the experience before they were asked to report it [201].

Another more mundane difficulty is that not all stimuli of the same general category are equal: For example, what is true of one-fold reflection need not be true of two-fold reflection. Statements such as "symmetry is processed automatically and pre-attentively" should be qualified by specifying the type of symmetry in question.

Despite these critical caveats, it is likely that *some kinds of symmetry are sometimes processed automatically, pre-attentively, and unconsciously, some of the time*. This conclusion relies on a range of evidence and arguments from multiple sources.

Based on SPN research, it is undeniable that high W reflectional symmetry in the retinal image is processed when it is not task-relevant. SPN research thus establishes task independence. However, it is possible that participants often spontaneously attend to symmetry in these experiments, even when it is irrelevant to their primary task. This seems implausible in some cases, but we cannot rule it out. Existing SPN research suggests, rather than establishes, that symmetry is often processed automatically, pre-attentively, and unconsciously.

The fact that symmetry can be detected quickly is again suggestive, but not conclusive. Symmetry is processed within 50 ms [68] and detected efficiently outside the foveal region [47]. This is perhaps indicative of pre-attentive symmetry processing. However, in a long experiment, participants may adopt a general mindset to search for symmetry across all trials. They may thus anticipate symmetry before stimulus onset. It is uncertain whether the symmetry would have been processed pre-attentively on a given trial without the participants adopting this symmetry-primed mindset (see [200] for related arguments regarding the interpretation of visual search tasks).

Some of the strongest evidence for automatic and pre-attentive symmetry processing comes from considerations about the role symmetry plays in perceptual organization. As described above, symmetry plays a role in figure-ground segmentation [13] and object identification [35]. The resulting visual representations guide thousands of saccades, actions, and verbal responses per day. This use of symmetry is likely to happen without spontaneous attention to symmetry itself.

Moreover, we can confidently say that symmetry can be processed unconsciously. Anesthetized animals are (hopefully) unconscious during invasive experiments. V4 cells are sensitive to different specific symmetrical patterns in anesthetized monkeys [126]. Furthermore, patient CC used symmetry in figure-ground segmentation, even though they could not discriminate symmetry consciously [132]. Morgan et al. [202] suggest imperfections can be discriminated in grids that look perfectly regular. In fact, this is not just unconscious, it cannot even be made conscious with prolonged introspective effort. Finally, sensitivity to symmetry in optic flow fields guides actions unconsciously when we walk around.

We can plausibly speculate that insects never become aware of visual dimensions in the same way as humans. For instance, they do not sub-vocally articulate remarks about their own visual experiences, such as 'look at that symmetrical flower'. However, even bees and spiders use visual symmetry to guide instinctive behavior [203]. Unconscious visual symmetry processing is thus a natural phenomenon, even if it does not happen in humans.

Despite these lines of argument, the literature includes some (hedged) claims that symmetry is NOT processed pre-attentively. Olivers and van der Helm [204] found that symmetry does not pop out in visual search tasks and concluded that "symmetry detection per se requires selective attention, but that some related grouping or segmentation mechanism may operate pre-attentively" (abstract). However, in visual search tasks, participants may only enter their responses after selective attention has scanned all items. Symmetry may have been processed pre-attentively in one location before this scanning op-

eration. The conclusions of Olivers and van der Helm [204] are consistent with the results of Kimchi et al. [205], who found that grouping by collinearity automatically captures spatial attention but grouping by symmetry does not. Most recently, Davytko and Kimchi [206] conclude that symmetry-based grouping cannot be accomplished in the absence of visual awareness. Based on the above, we suggest this claim does not generalize to perceptually stronger symmetries and symmetry-based grouping sometimes occurs without awareness.

Reflectional symmetry might be processed automatically, pre-attentively, and unconsciously because the visual brain is genetically hardwired to detect it. This possibility is discussed in part 4.

### 4. Part 4: Sensitivity to Reflectional Symmetry Is Innate

While genes do not hardwire the brain in excruciating detail, they certainly influence neural phenotype [207]. Perhaps sensitivity to reflectional symmetry is innate? Sensitivity to reflectional symmetry is evident in newly hatched poultry chicks [208], and insects use visual symmetry to guide instinctive behaviors [203]. This suggests hardwired symmetry detectors are a biological possibility. There is some evidence that humans have an innate sensitivity to reflectional symmetry. Preferential looking and habituation tasks suggest sensitivity to reflectional symmetry in 4-month-old human infants [209,210]. Indeed, Pornstein and Krinsky [211] found that habituation was stronger for vertical reflection than other regularities. They concluded that 6-month babies are uniquely sensitive to vertical reflection itself, not just verticality or reflection.

More recently, Ross-Sheehy et al. [212] found that only 6.5-month babies who had achieved self-supported sitting could use reflectional symmetry as a figural cue. They claimed that, while reflectional symmetry sensitivity is present at 4 months, babies only learn that symmetry is associated with whole objects once they can sit up and explore the world with their hands.

Connectionist simulations of visual discrimination have always informed the nature-nurture debate. Modern deep neural networks (DNNs), with multiple hidden layers, can be trained to recognize thousands of objects, without being directly programmed to do so [213]. DNN successes support a nurture position, and DNN failings indirectly support a nature position. For instance, Ridley claimed that failures of DNN stimulations ultimately show that *"Pre-programmed design is required for the solving of pre-ordained problems"* ([214] p. 103). This skepticism remains tenable, despite the huge advance in recent years [215–217].

Following earlier connectionist work [218], Sundaram et al. [219] trained several modern DNNs to recognize reflectional symmetry and then examined whether this learning generalized to test patterns with larger gaps between the reflected sides. Feedforward architectures were poor at finding reflection in untrained exemplars with novel gaps. One recurrent model, LSTM3, could generalize across gaps, but could not generalize between abstract and naturalistic images. The fact DNNs are insensitive to symmetry per se is consistent with other evidence that these networks struggle with gestalts. When classifying objects, networks rely on local features and are blind to the spatial relationships between them [216]. Alternative AI architectures, which win computer-vision symmetry detection competitions, apparently benefit from the pre-programmed design, although even these are not always as quick or accurate as humans [220]. This suggests that the pre-ordained problem of detecting reflectional symmetry requires pre-programmed neural design.

As mentioned, Pramod and Arun [128] found that pairs of symmetrical shapes are more perceptually and neurally distinct from each other than pairs of asymmetrical shapes. Pramod and Arun [221] expanded this with convolutional neural network (CNN) simulations. They measured perceptual dissimilarly between object pairs in humans and CNNs. Human and CNN dissimilarity scores were correlated, but not perfectly. CNNs tended to underestimate the dissimilarity between symmetrical object pairs. Augmenting CNNs with axis-of-symmetry information improved object classification performance. This suggests that some form of innate sensitivity to reflectional symmetry can boost perceptual learning.

As well as innate sensitivity to reflectional symmetry, we may also have an innate attraction to reflectional symmetry. However, there are many non-exclusive explanations for symmetry-philia, as discussed in part 5.

## 5. Part 5: Symmetry Is Aesthetically Significant

Many animals apparently use symmetry in mate selection [10,222,223]. This may be because phenotypic symmetry indicates genetic fitness. It is often noted that fluctuating asymmetry indicates developmental instability, disease or health problems. There may thus be a selection pressure that enhances symmetry sensitivity in brains, and a selection pressure that exaggerates symmetry in bodies. An alternative evolutionary model builds on the idea that animals need to be noticed by conspecifics. This leads to the evolution of bright, loud, distinct features. Symmetrical patterns are noticeable signals, partly because they look the same from various viewpoints [224,225]. Like many animals, humans are attracted to symmetrical faces [9,11] and bodies [226].

Empirical aesthetics is an old research field, beginning with the pioneering work of Gustav Fechner [227]. One of the oldest and most reliable findings is that people like symmetrical abstract patterns more than asymmetrical alternatives [228,229]. Symmetry is the strongest predictor of positive ratings [230], and symmetry preference can be reliably measured with implicit or explicit measures [231–233]. Non-accidental alignments and proportions, including symmetry, determine the aesthetic appeal of polygons, especially when these features are subtle [234]. However, artists and art experts may cultivate a taste for asymmetry [235]. It is unclear whether lay preference for symmetry in abstract patterns is an overgeneralization of innate preference for symmetry in faces and bodies, as briefly suggested by Ramachandran and Hirstein [12].

Symmetry implies an unseen force that has deliberately arranged the seen elements in a purposeful way. When contemplating symmetry, we may infer that someone or something has organized these elements, so they are not scattered chaotically. This echoes William James's famous definition of universal religious experience: *It consists of the belief that there is an unseen order and that our supreme good lies in harmoniously adjusting ourselves thereto* (opening sentence of Lecture 3, [236]). Artists and architects sometimes use symmetry to symbolize divine order. Enthusiasm about sacred geometry is also common in psychedelic literature, as discussed in part 6.

## 6. Part 6: Hallucinatory Symmetry

In part 6, we move to the more exotic topic of hallucinatory symmetry. We consider hallucinations induced by the psychedelic drug *N,N*-dimethyltryptamine (DMT), and by visual flicker. Symmetry is a highly prominent feature of both.

### 6.1. DMT

At low doses, DMT users commonly report a hallucinatory experience referred to as 'the chrysanthemum'. This public quote on a DMT web forum is representative:

> "Then I heard Terence [McKenna] mention that he closed his eyes and saw the Chrysanthemum, which I promptly did, and there it was. The most beautiful and perfect hallucination I could ever imagine having. A slowly rotating, fractal, kaleidoscopic, floral, mandalic, cone/tube/hemisphere on the back of my eyelids."

https://www.dmt-nexus.me/forum/default.aspx?g=posts&t=8595 (accessed on 28 June 2023).

At higher doses, the DMT experience becomes more surreal and immersive, with overflowing hyperdimensional symmetry often combined with the felt presence of supernatural entities. DMT trips are unimaginably intense aesthetic and spiritual experiences, that can never be captured by artistic impressions [237]. Many users report DMT experiences as life-changing, leading to radical and lasting shifts in their personal ontology. The well-known DMT advocate Terence McKenna said his psychedelic experiences are more terrifying, mind-blowing, and life-changing than, for example, switching on the TV news

to find that UFOs had landed on the White House lawn. He humorously claimed that DMT experiences are such an extraordinary departure from normal reality that users feel like they might 'die of astonishment' (Psychedelic Workshop 1992, recording available online, https://www.youtube.com/watch?v=mQ8t9o9d2Zg&t=14s, accessed on 28 June 2023)

DMT is from a family of serotonergic psychedelic drugs including LSD, mescaline, and psilocybin (magic mushrooms). Unlike LSD and psilocybin, DMT is an endogenous neurotransmitter, which has some signaling function in healthy brains. Individual differences in DMT signaling may contribute to individual differences in aesthetic sensitivity and religiosity, although this has not been demonstrated empirically.

While psychedelics are illegal in many countries and potentially dangerous, they might have some therapeutic benefits [238], although this is debatable [239]. The putative therapeutic value of psychedelics was discussed extensively in the 1950s and 60s, and research on psychedelics is now becoming mainstream again. Imperial College London now hosts the *Institute of Psychedelic Research* (https://www.imperial.ac.uk/psychedelic-research-centre/, accessed on 28 June 2023). In America, there is the more independent *Qualia Research Institute* (https://thequaliaresearchinstitute.org/, accessed on 28 June 2023).

The phenomenology of DMT experience is discussed by Lawrence et al. [14], who reported that hallucinatory symmetry and fractal geometry are common. A qualitative interview study by Cott and Rock [240] corroborates this. One participant reported that:

*"The entire room was crawling with beautiful geometric hallucinations."*

Another participant reported:

*"The room erupted in incredible neon colors, and dissolving into the most elaborate incredibly detailed fractal patterns that i [sic] have ever seen".*

The abundance of geometry is also obvious in DMT-inspired art (For instance, https://www.alexgrey.com/, accessed on 28 June 2023)). Weyl's claim that 'beauty is bound up with symmetry' [241] would sound like a dramatic understatement to DMT enthusiasts.

### 6.2. Six Stages of DMT Experience

A representative account from more mathematically minded DMT uses was compiled by Gomez-Emilsson [237]. At stage 1 (Threshold), colors become vivid and visual acuity subjectively sharpens. At stage 2 (The chrysanthemum), colorful slowly rotating kaleidoscopes and mandalas are salient when users close their eyes. The 17 wallpaper groups can be identified. If you keep your eyes open, surfaces 'symmetrify beyond recognition'. At stage 3 (The magic eye), surfaces may develop autostereogram properties, where depth separation breaks up the 2D surface. Any regular structure is prone to overflow and fractalize, to the point where if folds and warps into a hyperspace, with hyperbolic rather than mere Euclidian geometry. At Stage 4 (The waiting room), immersion hyperspace becomes intense, and there is a subjective sense of communicating with supernatural beings, either telepathically or in distorted language. The beings may be humanoid or insect, and distorted faces are common. At Stage 5 (The breakthrough), impossible hyper dimensionality and space-time tunneling are ubiquitous. The telepathic agents melt from the human form and are fused with the hyper-dimensional geometry. The new universe feels palpably panpsychic. Stage 6 (Amnesia) is apparently indescribable, because no visual representations or linguistic descriptions are adequate, and without these familiar cognitive tools, it is hard to reconstruct memories. While it is unlikely that these stages are universally experienced by all DMT users, the account again highlights the prominence of symmetry in psychedelic hallucinations.

### 6.3. Neural Research on Psychedelic Experience

Neuroimaging research on psychedelics is inconclusive, partly because studies are heterogeneous and underpowered [242]. Nevertheless, there has been progress in applying neurocomputational theories to psychedelic experience. Carhart-Harris and Friston [243] provide an account of DMT neurology and propose the Relaxed Beliefs Un-

der Psychedelics (REBUS) model, which is an application of hierarchical predictive coding [244,245]. Psychedelics are likely to alter communication across distributed neural networks. Schartner et al. [246] measured the algorithmic complexity of the spatio-temporal MEG signal, which correlates with the state of consciousness—complexity is low under anesthesia and sleeping, and high during waking conscious states. Interestingly, complexity was enhanced *beyond* the normal waking state with LSD, psilocybin, and low doses of ketamine.

Experimental work has examined changes in alpha rhythms under DMT. DMT reduces alpha power when the eyes are shut. Specifically, it reduces the power of backward traveling waves, which would inhibit the visual cortex [247]. Internally generated visual representations cascade forward to other parts of the brain, which then struggles to interpret them. It is intriguing that symmetry is such a prominent feature of these hallucinations.

### 6.4. Does DMT Enhance the SPN?

It has been shown that psychedelics reduce the mismatch negativity: an ERP response to deviant tones in a predictable sequence [248], and reduce the effect of coherent Kanizsa triangles on N1 [249]. However, nobody has conducted an experiment to determine whether DMT increases SPN amplitude. It may be difficult to conduct such an experiment. DMT might make all stimuli seem symmetrical, and the difference between symmetry and random waves might be masked. Furthermore, participants might stop fixating and attending to their tasks. However, there is some preliminary evidence that extrastriate activation scales with the degree of abstract geometry in the display. This abstraction dimension loosely mimics the change from normal visual experience to DMT-induced hallucination. Makin et al. [149] compared SPN responses in landscapes, flowers, and patterns, but did not interpret the much larger differences between these three classes. The results are shown in Figure 8. Negativity at posterior electrodes from 300 to 1000 ms is partly determined by symmetry (F (1, 19) = 56.266, $p < 0.001$, $\eta p^2 = 0.748$) but even more strongly determined by abstraction, as we go from realistic scenes to free-floating flowers, to abstract patterns (F (1.414, 26.857) = 75.988, $p < 0.001$, $\eta p^2 = 0.800$). This was replicated in three experiments, including two when people were not attending to symmetry. However, we caution that 'abstraction' is confounded with other visual dimensions in this study, so these results are not conclusive.

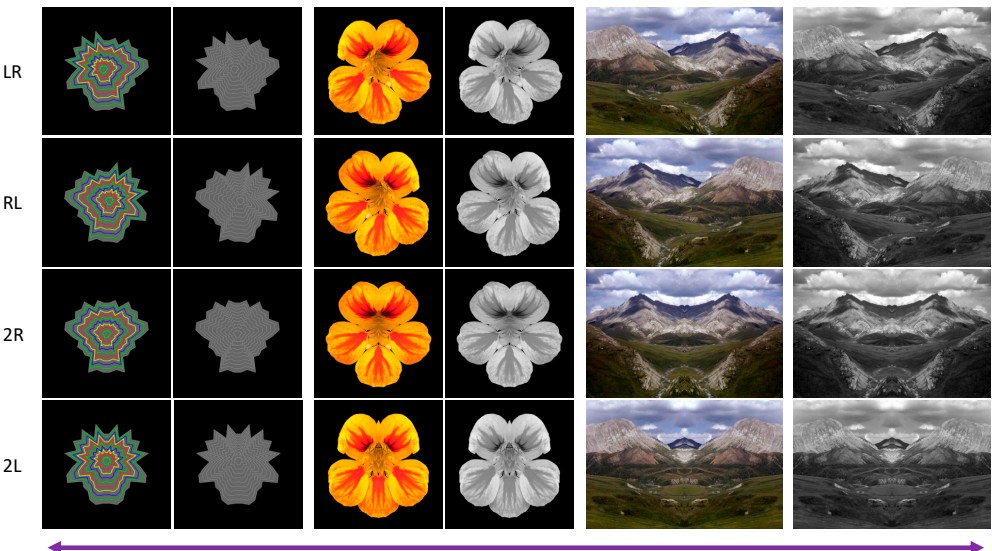

**Figure 8.** *Cont.*

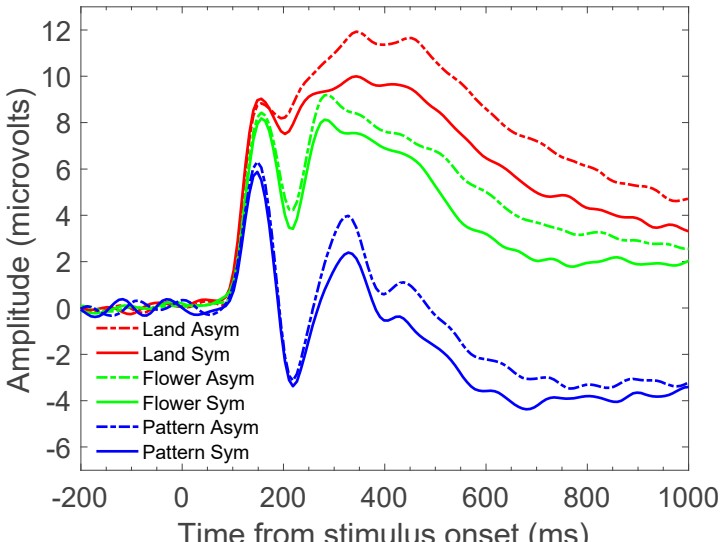

**Figure 8.** Results of Makin et al. [149]. Amplitude is more negative for symmetrical than asymmetrical stimuli. However, negativity is even more strongly determined by abstraction (Patterns < Floating Flowers < Landscapes).

*6.5. Flicker-Induced Hallucinations*

Early visual pioneers such as Purkinje and Fechner noticed that mild hallucinations involving form, color, and motion can be generated by unstructured flickering light [250]. Flicker-induced hallucinations can be categorized by Klüver's form constants, small set of geometries first identified as universal features of mescaline-induced hallucinations [251]. These are shown in Figure 9. As an aside, it can be noted that Glass patterns too can display radial, spiral, and concentric structures [252]. Klüver's form constants can be identified in pre-historic art [253] and modern artists have developed the *Dreammachine*, a recreational tool aimed at providing a safe, drug-free experience of visual hallucinations through visual flicker [254]. Interestingly, Ganzfeld-type sensory deprivation also leads to related hallucinatory experiences [255,256].

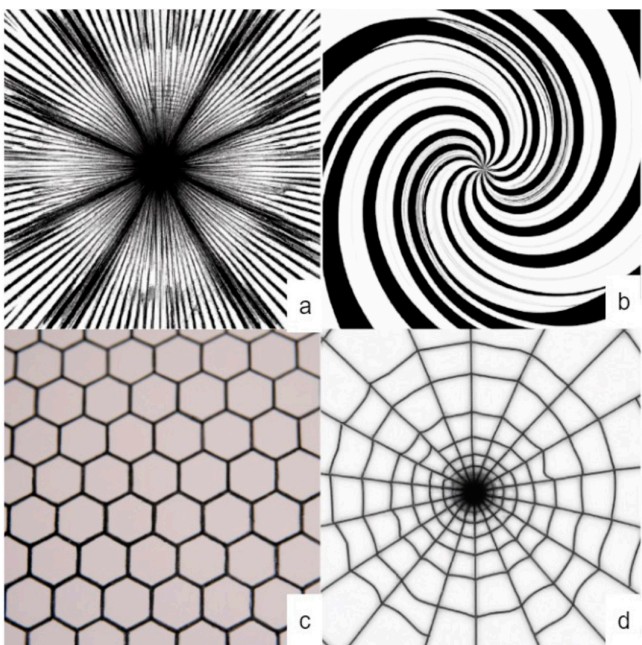

**Figure 9.** Artistic representations of form constants, redrawn from [15]. (**a**) tunnel/funnel; (**b**) spiral; (**c**) honeycomb; (**d**) cobweb.

Many experiments have systematically varied flicker frequency. Shevelev et al. [257] found frequencies in the alpha range were associated with ring, spiral, or grid-like hallucinations. There was a high correlation between participants' dominant alpha frequency and the optimal flicker frequency for reporting circular percepts. Hermann and Elliott [258] reported hallucinatory form percepts across the 5–39 Hz range, while Becker and Elliott [259] across the 8–40 Hz range. The most frequently reported subjective forms were variously symmetric, including circles, radial patterns, honeycombs, checkerboards, and spirals. Frequency-matched arrhythmic stimulation is less effective at inducing hallucinations [260]. Attempting to explain the link between stimulation frequency and the probability of perceiving specific geometries, Elliott et al. [261] proposed a mechanism based on the synchronization of receptive fields across cortical layers. More recently, Mauro et al. [262] advanced the hypothesis that cortical regions processing different geometries might be activated by different optimal stimulation frequencies.

To better characterize how flicker frequency affects subjective experience, there have been attempts to reduce the heterogeneity of hallucinatory percepts, for instance by employing a flickering annulus as opposed to an unstructured flickering field [263] or coupling flicker with static concentric rings or fan-shaped radial stimuli [264].

A seminal paper by Ermentrout and Cowan [265] put forward a modeling of the visual cortex as a two-layer neural net of excitatory and inhibitory neurons. Because of distinctive features of hallucinations, such as the fact that they can be experienced in darkness, they assumed that commonly hallucinated patterns are cortical in nature, as opposed to entoptic (e.g., because of ocular blood vessels becoming visible). Distinct percepts could be linked to the orientation of parallel activation stripes. Along the *x*-axis in the case of radial, fan-like percepts; along the *y*-axis in the case of concentric, tunnel-like percepts and along sloping lines in the case of spiral percepts. Follow-up work has shown how different form constants can be explained through more complex patterns of cortical activation in early areas [15,266–269]. More complex percepts may also be triggered by flicker, at least in people with high visual imagery abilities [270,271].

In summary, hallucinatory grids, fans, concentric circles, and spirals emerge when the early visual cortex is overstimulated by flickering light. There may be a mechanistic overlap between flicker hallucinations and some low-dose DMT-induced hallucinations, such as the chrysanthemum [272].

## 7. Conclusions

A review of the psychophysical literature (part 1) suggests filter models account for some, but not all aspects of symmetry perception. Symmetry in optic flow fields and radial frequency contours might be a different category. A review of the neuroscientific literature (part 2) shows that the extrastriate cortex is sensitive to symmetry, and the brain responds to symmetry in the image even when it is not task-relevant. Indeed, some forms of symmetry are processed automatically, pre-attentively, and unconsciously (part 3). Furthermore, sensitivity to reflectional symmetry may be innate (part 4). Symmetry is an important aspect of aesthetic experience (part 5). Finally, symmetry is prominent in hallucinations produced by exposure to visual flicker, and dramatically prominent in hallucinations induced by DMT (part 6).

The six parts of this review are modular, and they can be consulted independently. However, research in each area will be informed by the others. First, anything that enhances symmetry detection in psychophysical studies is likely to enhance the extrastriate symmetry response and aesthetic appreciation. Second, reflectional symmetry is special for the visual system, and this may partly reflect innate sensitivity. Third, researchers working on the neural response to symmetry should be aware that symmetry is a prominent feature of drug and flicker-induced hallucinations.

The psychedelic renaissance opens new doors for symmetry perception research. We know the extrastriate cortex responds to symmetry in the world—but what would the extrastriate cortex do if there was no input from sensory stimuli and no top-down inhibition

from other brain areas? Perhaps it would not simply remain silent by default. Instead, it may spontaneously generate a kaleidoscope of colorful and geometrical representations. Psychedelic drugs may disinhibit the extrastriate cortex, so the internally generated kaleidoscope intrudes on conscious experience.

**Author Contributions:** A.D.J.M. conceptualized the review and wrote the first draft. M.R. added a section on Flicker-induced hallucinations (part 6). E.K. contributed to the review of perspective symmetry in part 1. J.T.-C. contributed to the review of neuroscientific work in part 2. M.B. helped with the overall structure and commentary. All authors have read and agreed to the published version of the manuscript.

**Funding:** Makin was funded by the Economic and Social Research Council Grant ESRC grant while writing this review (ES/S014691/1).

**Data Availability Statement:** We encourage readers to visit our SPN catalog on Open Science Framework if they wish to examine the EEG database discussed in part 2 (https://osf.io/2sncj/, accessed on 28 June 2023).

**Conflicts of Interest:** The authors declare no conflict of interest.

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
