# Peer review of "Symmetry Perception and Psychedelic Experience"

_symmetry, doi:10.3390/sym15071340_

Round 1
Reviewer 1 Report
The review provides a comprehensive overview of the research on visual symmetry perception, integrating findings from psychophysics, neuroscience, aesthetics, and hallucinatory experiences. It highlights the limitations of filter models in explaining certain aspects of symmetry perception and emphasizes the need for a more comprehensive understanding of symmetry processing in the brain.
The abstract provides an overview of the main points discussed in the article. It states that recent research has revealed the involvement of the extrastriate visual cortex in processing visual symmetry. The review aims to integrate neuroscientific findings on symmetry processing with research on hallucinations induced by the drug DMT and visual flicker. The abstract also mentions the importance of symmetry in aesthetics and its role in image segmentation. The keywords indicate the focus areas of the article.
However, there are a few areas where the abstract could be improved. Firstly, it would be beneficial to provide more specific details about the recent neuroscientific findings related to symmetry processing in the extrastriate cortex. Additionally, the abstract mentions the integration of psychedelic experiences induced by DMT and visual flicker with symmetry processing, but it does not clearly explain the significance or implications of this integration. Providing more information on this aspect would make the abstract more informative and engaging for the reader.
In terms of the language and structure, the review is generally well-written. The sentences are clear and concise, and the abstract flows logically from one topic to another. The abstract also includes relevant keywords that accurately represent the main themes of the article.
The introduction provides a brief background on the study of visual symmetry perception, highlighting its long history in psychophysical research. It mentions the relevance of symmetry in various domains, such as sexual attraction, art, aesthetics, and hallucinations induced by DMT and visual flicker. The introduction outlines the structure of the review, which includes different parts focusing on psychophysical work, brain responses to symmetry, automatic processing, aesthetic significance, and hallucinations.
The conclusion should provide a concise summary of the main findings discussed in the preceding sections, rather than simply stating the part numbers. This will help the reader to better understand the key takeaways of the review.
The review is too long. If the authors were to shorten the review, it would greatly facilitate its readability.
Author Response
The review provides a comprehensive overview of the research on visual symmetry perception, integrating findings from psychophysics, neuroscience, aesthetics, and hallucinatory experiences. It highlights the limitations of filter models in explaining certain aspects of symmetry perception and emphasizes the need for a more comprehensive understanding of symmetry processing in the brain.
The abstract provides an overview of the main points discussed in the article. It states that recent research has revealed the involvement of the extrastriate visual cortex in processing visual symmetry. The review aims to integrate neuroscientific findings on symmetry processing with research on hallucinations induced by the drug DMT and visual flicker. The abstract also mentions the importance of symmetry in aesthetics and its role in image segmentation. The keywords indicate the focus areas of the article.
However, there are a few areas where the abstract could be improved. Firstly, it would be beneficial to provide more specific details about the recent neuroscientific findings related to symmetry processing in the extrastriate cortex. Additionally, the abstract mentions the integration of psychedelic experiences induced by DMT and visual flicker with symmetry processing, but it does not clearly explain the significance or implications of this integration. Providing more information on this aspect would make the abstract more informative and engaging for the reader.
We have now made these improvements to the abstract:
“Abstract: This review of symmetry perception has 6 parts. Psychophysical studies have investigated symmetry perception for over 100 years (part 1). Neuroscientific studies on symmetry perception have accumulated in the last 20 years. Functional MRI and EEG experiments have conclusively shown that regular visual arrangements, such as reflectional symmetry, Glass patterns, and the 17 wallpaper groups all activate the extrastriate visual cortex. This activation generates an event related potential (ERP) called the sustained posterior negativity (SPN). SPN amplitude scales with the degree of regularity in the display, and the SPN is generated whether participants attend to symmetry or not (part 2). It is likely that some forms of symmetry are detected automatically, unconsciously, and preattentively (part 3). It might be that the brain is hardwired to detect reflectional symmetry (part 4), and this could contribute to its aesthetic appeal (part 5). Visual symmetry and fractal geometry are prominent in hallucinations induced by the psychedelic drug N, N-dimethyltryptamine (DMT) and visual flicker (part 6). Integrating what we know about symmetry processing with features of induced hallucinations is a new frontier in neuroscience. We propose that the extrastriate cortex can generate aesthetically fascinating symmetrical representations spontaneously, in the absence external symmetrical stimuli. “
In terms of the language and structure, the review is generally well-written. The sentences are clear and concise, and the abstract flows logically from one topic to another. The abstract also includes relevant keywords that accurately represent the main themes of the article.
The introduction provides a brief background on the study of visual symmetry perception, highlighting its long history in psychophysical research. It mentions the relevance of symmetry in various domains, such as sexual attraction, art, aesthetics, and hallucinations induced by DMT and visual flicker. The introduction outlines the structure of the review, which includes different parts focusing on psychophysical work, brain responses to symmetry, automatic processing, aesthetic significance, and hallucinations.
The conclusion should provide a concise summary of the main findings discussed in the preceding sections, rather than simply stating the part numbers. This will help the reader to better understand the key takeaways of the review.
We have now included a new summary paragraph in the conclusion section:
“The six parts of this review are modular, and they can be consulted independently. However, research in each area will be informed by the others. First, anything that enhances symmetry detection in psychophysical studies is likely to enhance the extrastriate symmetry response and aesthetic appreciation. Second, reflectional symmetry is special for the visual system, and this may partly reflect innate sensitivity. Third, researchers working on the neural response to symmetry should be aware the symmetry is a prominent feature of drug and flicker induced hallucinations.”
The review is too long. If the authors were to shorten the review, it would greatly facilitate its readability.
While it is important to keep things as concise as possible, another aspiration of a review paper is to cover all relevant literature. We feel a good review paper would be useful for a postgraduate student who is planning to write a new thesis on symmetry perception. We want them to think ‘oh, great, it’s all in here!’.
However, we have shortened in places:
We have removed the section on regular grids to shorten the review.
We have removed a paragraph about axis scanning from the bootstrapping section.
We have removed a paragraph about technicalities of the Daytko and Kimchi paper.
We have removed a paragraph about perceptual fluency.
Reviewer 2 Report
The manuscript is a review about perception of symmetry divided into different sections that are claimed to be understandable in isolation.
The review is interesting for, precisely, reviewing to much literatura about the topic. However, there are some things that I think should be improved:
#1. In the beginning of the work, there is no research question or hypothesis or goals. I think authors should introduce the topic in a more scientific way, and then, along the text, approach those goals through the published science. Otherwise, it is just a briefing of previous experiments with no added value.
#2. There is no connection among the sections.
#3. There is no explanation of many types of symmetry and authors suppose the reader already knows them when they should better describe them all at the beginning. A couple of examples would be 2D planar symmetry or retinal symmetry. But note that this is for all types of symmetries explained here.
#4. Figure 1 has texts in red.
#5. I understand authors have managed the rights to put so many figures from other publications.
#6. Page 7, authors cite Bertamini et al. (2018) in APA instead of the journal's citation style.
#7. Authors make non-rigorous claims that, I think, should be avoided. Example: page 13 "we believe that retinal...", I think it should be more an hypothesis.
#8. The conclusion should connect all the parts to give answer to specific questions.
Author Response
The manuscript is a review about perception of symmetry divided into different sections that are claimed to be understandable in isolation.
The review is interesting for, precisely, reviewing to much literatura about the topic. However, there are some things that I think should be improved:
#1. In the beginning of the work, there is no research question or hypothesis or goals. I think authors should introduce the topic in a more scientific way, and then, along the text, approach those goals through the published science. Otherwise, it is just a briefing of previous experiments with no added value.
While we accept that brevity is desirable in many contexts, we would do not entirely agree with this criticism. Review articles would not conventionally be centred on a hypothesis, in the way a research paper would be. A briefing of previous experiments is exactly what some readers require.
Moreover, we do far more than just list previous experiments with no added value. Amongst other things. we consider the limitations of filter models, we argue that sensitivity to reflectional symmetry might be innate, we draw conclusions from analysis of the whole SPN catalogue. Most importantly, we highlight the prominence of symmetry in DMT and flicker induced hallucinations. This is an important consideration for those working the neuroscience of symmetry, and a gap in the literature.
#2. There is no connection among the sections.
We have now included specific signposts linking between the sections.
“In part 2, we consider the brain response to visual symmetry. Most things that enhance symmetry detection in psychophysical tasks also enhance the neural response.”
“Many non-accidental visual configurations activate the shape-sensitive LOC, and other parts of the extrastriate cortex. Symmetrical arrangements produce strong responses. The extrastriate symmetry scales with the goodness of symmetry in the image, as shown with fMRI, the SPN, SSVEPs and alpha ERD. Anything that alters symmetry discrimination performance is likely to alter SPN amplitude. There is strong evidence that retinal symmetry activates the extrastriate cortex automatically. However, this requires careful consideration in part 3.
“Reflectional symmetry might be processed automatically, pre-attentively and unconsciously because the visual brain is genetically hardwired to detect it. This possibility is discussed in part 4.
“As well as innate visual sensitivity to symmetry, we may also have an innate attraction to reflectional symmetry. However, there are many non-exclusive explanations for symmetry-philia, as discussed in part 5.”
“Artists and architects sometimes use symmetry to symbolize divine order. Enthusiasm about sacred geometry is also common in psychedelic literature, as discussed in part 6.”
#3. There is no explanation of many types of symmetry and authors suppose the reader already knows them when they should better describe them all at the beginning. A couple of examples would be 2D planar symmetry or retinal symmetry. But note that this is for all types of symmetries explained here.
We have now explained this on page 1:
“When people use the word ‘symmetry’ they usually think of reflectional, mirror symmetry. Many papers use the word symmetry as a synonym for reflection, because that is the only type investigated. This review covers other types of symmetry and regularity as well..”
This now links to the next section, where we said:
“An arrangement is symmetrical if it remains identical after rigid transformation. The 2D planar symmetries are reflection, glide reflection, rotation, and translation (also referred to as repetition). The famous physicist Ernst Mach [16] noticed that reflectional symmetry is more salient than rotation or translation, even when it has the same number of rigid transformations (Figure 1A).”
We have also added new examples to Figure 1A
#4. Figure 1 has texts in red.
We are unsure about the colour rules for the figures, but we have now changed to purple and black text.
#5. I understand authors have managed the rights to put so many figures from other publications.
Yes. We have now clarified this in the Figure legends.
#6. Page 7, authors cite Bertamini et al. (2018) in APA instead of the journal's citation style.
Fixed
#7. Authors make non-rigorous claims that, I think, should be avoided. Example: page 13 "we believe that retinal...", I think it should be more an hypothesis.
We have replaced phrases like ‘we believe’ with less personal equivalents such as ‘the evidence suggests’.
#8. The conclusion should connect all the parts to give answer to specific questions.
We have added a new paragraph connecting the parts:
“The six parts of this review are modular, and they can be consulted independently. However, research in each area will be informed by the others. First, anything that enhances symmetry detection in psychophysical studies is likely to enhance the extrastriate symmetry response and aesthetic appreciation. Second, reflectional symmetry is special for the visual system, and this may partly reflect innate sensitivity. Third, researchers working on the neural response to symmetry should be aware the symmetry is a prominent feature of drug and flicker induced hallucinations.”
Reviewer 3 Report
Literature Review on the perception of symmetry.
Written in correct language, well-structured and very welcoming to interdisciplinary scientists
The fact that low freqeuncies dominate high frequencies in symmetry perception can stimulate interdisciplinary research, deriving from the field on fMDRI (which seens to be the author´s specialty)
and extending towards linear signal processing (such as electrophysiological recordings) or image
processing (radiology, pathology). Being an appasionate comparative neuroscientist, I inguinely
enjoyed excellent references to comparative neuroscience concerning for example the avian visual
system (L92).
A remarkable reference on the symptom of contralateral neglect (L303-306) upon the parietal cortex lesions, which is still confused with hemianopsia and often blurs the accurate clinical diagnosis of a stroke.
Reviewer´s suggestions:
001 - It is suggested to include the term comparative neuroscience in your keyword list (see L92-reference to the avian symmetry perception).
002 - L18 – Bring up the aspect of aesthetics in the abstract to better substantiate the connection between the keywords “aesthetics” and “nature-nurture” with the scientific topic of this article
003 – L33, L77 – correct to “flicker-induced” and “low-pass”; please proof-check your text for hyphens
004 - An interesting aspect would be to show some studies investigating how much of non-symmetrical high-frequency noise one has to inject in order to disrupt the low-frequency symmetry perception and vice versa. Immense importance of such studies might be revealed in the field of psychiatric research, or even in the field of image psychology, advertisement and so on.
005 – elaborate on how symmetric patterns can initiate epileptic discharges
Author Response
Literature Review on the perception of symmetry.
Written in correct language, well-structured and very welcoming to interdisciplinary scientists
The fact that low freqeuncies dominate high frequencies in symmetry perception can stimulate interdisciplinary research, deriving from the field on fMDRI (which seens to be the author´s specialty) and extending towards linear signal processing (such as electrophysiological recordings) or image processing (radiology, pathology). Being an appasionate comparative neuroscientist, I inguinely enjoyed excellent references to comparative neuroscience concerning for example the avian visual system (L92).
A remarkable reference on the symptom of contralateral neglect (L303-306) upon the parietal cortex lesions, which is still confused with hemianopsia and often blurs the accurate clinical diagnosis of a stroke.
We are pleased that reviewer 3 is positive about our review article.
Reviewer´s suggestions:
001 - It is suggested to include the term comparative neuroscience in your keyword list (see L92-reference to the avian symmetry perception).
We have done this (although we note that there is just one sentence on the avian visual system). We have also added more about preference for symmetry in animals:
“Many animals apparently use symmetry in mate selection [10,222,223]. This may be because phenotypic symmetry indicates genetic fitness. It is often noted that fluctuating asymmetry indicates developmental instability, or disease or health problems. There may thus be a selection pressure which enhances symmetry sensitivity in brains, and a selection pressure that exaggerates symmetry in bodies. An alternative evolutionary model builds on the idea that animals need to be noticed by conspecifics. This leads to evolution of bright, loud, distinct features. Symmetrical patterns are noticeable signals, partly because they look the same from various viewpoints [224,225]. Like many animals, humans are attracted to symmetrical faces [9,11] and bodies [226].”
002 - L18 – Bring up the aspect of aesthetics in the abstract to better substantiate the connection between the keywords “aesthetics” and “nature-nurture” with the scientific topic of this article
We have done this.
“Abstract: This review of symmetry perception has 6 parts. Psychophysical studies have investigated symmetry perception for over 100 years (part 1). Neuroscientific studies on symmetry perception have accumulated in the last 20 years. Functional MRI and EEG experiments have conclusively shown that regular visual arrangements, such as reflectional symmetry, Glass patterns, and the 17 wallpaper groups all activate the extrastriate visual cortex. This activation generates an event related potential (ERP) called the sustained posterior negativity (SPN). SPN amplitude scales with the degree of regularity in the display, and the SPN is generated whether participants attend to symmetry or not (part 2). It is likely that some forms of symmetry are detected automatically, unconsciously, and preattentively (part 3). It might be that the brain is hardwired to detect reflectional symmetry (part 4), and this could contribute to its aesthetic appeal (part 5). Visual symmetry and fractal geometry are prominent in hallucinations induced by the psychedelic drug N, N-dimethyltryptamine (DMT) and visual flicker (part 6). Integrating what we know about symmetry processing with features of induced hallucinations is a new frontier in neuroscience. We propose that the extrastriate cortex can generate aesthetically fascinating symmetrical representations spontaneously, in the absence external symmetrical stimuli. “
003 – L33, L77 – correct to “flicker-induced” and “low-pass”; please proof-check your text for hyphens
Done
004 - An interesting aspect would be to show some studies investigating how much of non-symmetrical high-frequency noise one has to inject in order to disrupt the low-frequency symmetry perception and vice versa. Immense importance of such studies might be revealed in the field of psychiatric research, or even in the field of image psychology, advertisement and so on.
We are unclear how this would be of immense importance in psychiatric research. However, we agree that this in an interesting topic. Indeed, it was an aspect of the research by Julesz and Chang (1979). We have now expanded on this:
“One early study using this approach found that when horizontal and vertical reflections with similar frequencies are superimposed, the result appears random. However, when their spatial frequencies are more than two octaves apart, they become perceptually separated, although the low band array tends to dominate [Figure 1C, 39].”
005 – elaborate on how symmetric patterns can initiate epileptic discharges
We are not sure this is true. If the reviewers would like to provide a reference on symmetry and epileptic discharges, we would incorporate it.
Photosensitive epilepsy is usually triggered by very high contrast or moving patterns. The kinds of visual flicker discussed in the flicker-induced hallucination section might trigger epilepsy. However, as far as well know, static symmetric patterns do not.
Round 2
Reviewer 2 Report
Thanks for approaching all my comments.